# Advanced Glycation End Product (AGE) and Soluble Receptor of AGE (sRAGE) Levels in Relation to Periodontitis Severity and as Putative 3-Year Outcome Predictors in Patients Undergoing Coronary Artery Bypass Grafting (CABG)

**DOI:** 10.3390/jcm11144105

**Published:** 2022-07-15

**Authors:** Stefan Reichert, Britt Hofmann, Michael Kohnert, Alexander Navarrete Santos, Lisa Friebe, Julia Grollmitz, Hans-Günter Schaller, Susanne Schulz

**Affiliations:** 1Department of Operative Dentistry and Periodontology, Martin-Luther-University Halle-Wittenberg, 06112 Halle, Germany; lisa.friebe@hotmail.de (L.F.); julia.grollmitz@uk-halle.de (J.G.); hans.guenter.schaller@uk-halle.de (H.-G.S.); susanne.schulz@medizin.uni-halle.de (S.S.); 2Department of Cardiac Surgery, Mid-German Heart Centre of the University Hospital Halle (Saale), Martin-Luther-University Halle-Wittenberg, 06120 Halle, Germany; britt.hofmann@uk-halle.de (B.H.); michael_kohnert@gmx.de (M.K.); 3Centre for Medical Basic Research, Martin-Luther-University Halle-Wittenberg, 06120 Halle, Germany; alexander.navarrete@uk-halle.de

**Keywords:** periodontitis, CABG surgery, cardiovascular outcome, sRAGE, AGE, skin autofluorescence, sAF

## Abstract

Tissue concentrations of advanced glycation end product (AGE) and peripheral soluble receptor of AGE (sRAGE) levels may be associated with periodontitis severity. Both parameters and periodontitis might serve as outcome predictors for patients undergoing coronary artery bypass grafting (CABG). This study aimed to investigate possible associations between periodontitis and AGE/sRAGE. Ultimately, we wanted to examine whether AGE, sRAGE, and severe periodontitis are associated with the incidence of new cardiovascular events within 3 years of follow-up after CABG. Ninety-five patients with coronary vascular disease (CVD) (age 69 years, 88.3% males) needing CABG surgery were included. Periodontal diagnosis was made according to the guidelines of the “Centers for Disease Control and Prevention (CDC)” (2007) and staged according to the new classification of periodontal diseases (2018). AGE tissue concentrations were assessed as skin autofluorescence (sAF). sRAGE levels were determined by using a commercially available enzyme-linked immunoabsorbance assay (ELISA) kit. Univariate and multivariate baseline and survival analyses were carried out with Mann–Whitney U test, Chi² test, Kaplan–Meier curves with Log-Rank test, and logistic and Cox regression. sAF was identified as an independent risk indicator for severe periodontitis with respect to the cofactors age, gender, plaque index, and diabetes (adjusted odds ratio [OR] = 2.9, *p* = 0.028). The degree of subgingival inflammation assessed as a percentage of sites with bleeding on probing (BOP) was inversely correlated with sRAGE concentration (r = −0.189, *p* = 0.034). Both sAF (Hazard Ratio [HR] = 2.4, *p* = 0.004) and sRAGE (HR = 1.9, *p* = 0.031) increased the crude risk for new adverse events after CABG. The occurrence of severe periodontitis trends towards a higher risk for new cardiovascular events (HR = 1.8, *p* = 0.115). Applying multivariate Cox regression, only peripheral arterial disease (adjusted HR = 2.7, *p* = 0.006) and history of myocardial infarction (adjusted HR = 2.8, *p* = 0.010) proved to be independent risk factors for cardiovascular outcome. We conclude that sAF may represent a new, independent risk indicator for severe periodontitis. In contrast, sAF, sRAGE, and severe periodontitis were not independent prognostic factors for postoperative outcome in patients undergoing CABG.

## 1. Introduction

Coronary artery bypass grafting (CABG) surgery is a standard, proven cardiac procedure for patients with coronary multi-vessel disease and/or left main coronary stenosis. CABG survival rates depend, among other factors, on age [1], emergency operation, shock, preoperative renal failure, longer total bypass time, intraoperative stroke, postoperative myocardial infarction (MI), gastrointestinal complications, respiratory failure [2], diabetes [3], and peripheral arterial disease (PAD) [4]. 

A large number of studies suggest that there is an independent association between periodontitis and cardiovascular disease (CVD) [5]. Therefore, it is conceivable that periodontitis also influences the postoperative course after CABG surgery. Could such an association be shown, periodontal therapy would be indicated prior to elective heart surgical procedures. Recently, in a case–control study, it was demonstrated that periodontal therapy before cardiac valve surgery reduced the number of postoperative days with a high fever (>37.5 °C). The authors concluded that periodontal treatment might contribute to reducing the risk of a postoperative infection [6]. In an earlier study of our group, the incidence of new adverse events tended to increase after CABG among patients with severe periodontitis. However, this relationship was not significant, probably due to the short follow-up time of 1 year. A renewed examination of patients after a longer follow-up period would therefore be useful [7]. 

In recent years, various biomarkers have been identified that could play a role in the pathology of both CVD and periodontitis. Advanced glycation end products (AGEs) represent one such biomarker. AGEs arise due to the Maillard reaction. The first step is the nonenzymatic binding of reduced sugars to free amino groups of proteins, lipids, and nucleic acids. This leads to the formation of unstable compounds known as Schiff bases. After several days, more stable Amadori products such as the hemoglobin A1c (HbA1c) develop. Over a period of weeks and months and through a series of oxidation reactions, stable AGEs arise [8]. In addition to the endogenous formation of AGEs, these can also be absorbed externally via digestion [9] or by inhaling tobacco smoke [10]. 

AGEs are involved in the pathogenesis of atherosclerosis via two mechanisms: (i) directly, altering the functional properties of vessel wall or extracellular matrix molecules, leading to artery stiffness, or (ii) indirectly, through interaction with the receptor for AGEs (RAGE), a transmembrane signaling receptor that is present in all cells relevant to atherosclerosis. As a consequence, for instance, the gene expression and release of proinflammatory cytokines can increase [11]. 

AGE may also be important in the pathogenesis of periodontitis. Levels of AGE in gingival crevicular fluid (GCF) were found to be higher in patients with type 2 diabetes mellitus and chronic periodontitis than in systemically healthy individuals with and without periodontitis [12]. In the gingiva of diabetic mice, more AGE and subsequent expression of RAGE were detected. In gingival fibroblasts, AGEs induce nuclear factor-kappa B, leading to increased production of interleukin-6 and tumor necrosis factor (TNF). Moreover, accumulation of AGEs in bone stimulates apoptosis of osteoblasts and interferes with bone regeneration. In contrast, if RAGE was inhibited in diabetic mice, the production of TNF was reduced and less periodontal bone loss was shown (overview [13]). 

Specific AGEs exhibit intrinsic fluorescence properties. Thus, AGE tissue burden can be estimated noninvasively by measuring the skin autofluorescence (sAF) [14]. 

Soluble isoforms of RAGE (sRAGE) may constitute further new and interesting biomarkers for CABG outcome and severe periodontitis. As the sRAGE forms lack the transmembrane and cytosolic domain, subsequent downstream signaling is not possible. Therefore, sRAGE is thought to act as a competitive inhibitor of AGE-RAGE interaction and of subsequent inflammatory reactions that are important in the pathogenesis of atherosclerosis and CVD [15]. The results of studies on humans investigating a putative association between sRAGE serum levels and atherosclerosis and subsequent diseases such as CVD and cerebrovascular disease were inconsistent. Indeed, some studies reported inverse associations between the two parameters whereas others did not (overview [16]). In the context of CABG, the results of two studies suggest that sRAGE levels are putative predictors for postoperative outcome. One study revealed that sRAGE measured after CABG was associated with a prolonged postoperative respiration time/stay in the intensive care unit or with the need for catechelamine support [17]. Another study found an association between postoperative sRAGE levels and prolonged length of hospitalization in patients after CABG [18]. 

Only a few studies have investigated the association between sRAGE and periodontitis with consistent results. sRAGE levels in serum and GCF were found to be reduced in patients with periodontitis and inversely correlated with the number of sites with bleeding upon probing [19,20,21]. Apparently, sRAGE seems to inhibit the inflammatory reaction on the microbial plaque by binding potential AGE ligands in periodontitis.

We hypothesized that severe periodontitis is associated with both altered sRAGE levels and altered sAF. Periodontitis itself, sRAGE, and sAF might be putative predictors for CABG outcome. The following objectives of the study were derived from this hypothesis. First, it should be investigated in patients undergoing CABG surgery whether the degree of severity of periodontitis was associated with preoperatively determined sAF and sRAGE concentrations; the second aim of our study was to investigate whether the severity of periodontitis and sAF and sRAGE levels are associated with cardiovascular outcome after CABG surgery within a 3-year follow-up period. 

## 2. Materials and Methods

### 2.1. Patient Enrollment 

The present work was a substudy of a trial entitled “Oral microbiome as a predictor for new cardiovascular events in patients with coronary heart disease”. The study is registered in the “German Register for Clinical Studies (DRKS) ID: DRKS00015776. We conducted a longitudinal cohort analysis with a follow-up period of 3 years. Out of a population of 308 patients with CVD for whom CABG surgery was indicated, 102 were ultimately enrolled at the Department of Cardiac Surgery of the Mid-German Heart Center at the University Hospital Halle (Saale) between January and October 2017. An experienced cardiac surgeon (B.H.) as well as two clinically calibrated dentists (L.F. and J.G.) diagnosed all patients regarding cardiological and periodontal conditions and assessed all patients for study eligibility. The following inclusion criteria had to be met: Caucasian descent, age ≥ 18 years, at least 60% stenosis of one of the main coronary arteries demonstrated by angiography, and the presence of at least four teeth. Exclusion criteria were inability to give written informed consent, subgingival scaling and root planing and/or antibiotic therapy during the last 6 months prior to the examination, pregnancy, and the need of antibiotic prophylaxis against endocarditis according to the criteria of the European Society for Cardiology [22]. Moreover, patients with diseases or disorders such as current drug or alcohol abuse that preclude participation in this clinical study according to investigator judgment were excluded. In all, 206 patients were not included in the study because they did not meet the inclusion criteria, a dental examination was not possible before the CABG surgery, a cardiac emergency existed, or the patients did not consent to participate in the study. After an observation period of over 3 years (mean 148 + 8.3 weeks), data on the postoperative cardiovascular outcome of 95 patients (drop out: 9.3%) could be generated. The study flow is shown in Figure 1. 

### 2.2. Demographic Parameters and Clinical and Cardiological Diagnostics

Baseline variables such as age, smoking status (never, past, and current smokers), and current or past diseases (e.g., diabetes mellitus, hypertension, PAD, and dyslipoproteinemia) were assessed as part of the patient’s medical history. A person who smoked a minimum of one cigarette per day at the time of questioning was considered to be a current smoker. A past smoker had not smoked for at least 1 year at the time of the survey. Furthermore, all patients underwent detailed clinical and biochemical investigation. For instance, intake of drugs such as lipid-lowering drugs, oral anticoagulants, and antiarrhythmics were registered. Serum parameters, including creatinine [µmol/L], urea [mmol/L], glycated hemoglobin (HbA1c) [mmol/mol], C-reactive protein (CRP) [mg/dL], leukocytes [Gpt/L], platelets [Gpt/L], triglycerides (mmol/L), and glucose (mmol/L), were recorded. In order to assess the severity of the CVD, the number of coronaries affected was determined (one-, two-, or three-vessel disease) as was the left ventricular ejection fraction (LVEV, %).

### 2.3. Dental Anamnesis and Examinations

The dental anamnesis and examination were conducted 1 day before the CABG surgery. In order to reduce the risk of bacteremia due to the probing of dental pockets, the participants were asked to rinse their mouth with an antibacterial wash solution (* Chlorhexamed^®^ FORTE alcohol-free 0.2%, GlaxoSmithKline Consumer Healthcare GmbH & Co. KG, Munich, Germany) before the dental examination. First, the number of missing teeth was registered. The clinical dental assessment involved determining the plaque index (PI) [23] and bleeding on probing (BOP) [24]. In the plaque index, four tooth surfaces were evaluated: mesio-buccal, mid-buccal, disto-bucca, and lingual. In the bleeding index, six sites around each tooth (mesio-buccal, mid-buccal, disto-buccal, disto-oral, mid-oral, and mesio-oral) were examined. BOP was only evaluated after a waiting time of 30 s after probing. Furthermore, the measurements for both maximal clinical probing depth (PD = distance between gingival margin and the apical stop of the probe) and maximum clinical attachment loss (CAL = distance between the cementoenamel junction and the apical stop of the probe) were taken also at six sites around each tooth. The maximum values for each tooth were used to calculate the overall mean per participant. In order to obtain reproducible results for BOP, PD, and CAL, the two examiners (L.F. and J.G.) were particularly trained in using a pressure-sensitive dental probe (UNC 15 0.2 N Aesculap, Tuttlingen, Germany) and clinically calibrated. The examiners paid particular attention to orient the probe in the direction of the tooth axis. The reading was made exactly to a millimeter. If one measuring point (gingival margin or cementoenamel junction) was between two markers of the measuring scale, the measurement was estimated to be 0.5 mm. For the clinically calibration, both examiners determined PD and CAL twice on five periodontal phantom models (phantom model A-PB, frasaco GmbH, Tettnang, Germany) and on five patients. To assess the reproducibility of the double measurements, the Bland–Altman method was used [25]. The difference (d) between the two measurements was calculated and plotted against the mean of the two measurements. The measurements were sufficiently reproducible if 95% of the differences (d) were in the range d ± 2 × s, where s denotes the standard deviation of the differences. Regarding our two raters, the differences between the two measurements for PD and CAL were found to be 100% in the range d ± 2 × s. Thus, the examiners L.F. and J.G. were able to generate reproducible measurements.

A clinical periodontitis case was defined according to the guidelines of the working group of the Centers for Disease Control and Prevention (CDC) [26]. According to CDC, severe periodontitis was diagnosed if at least ≥2 interproximal sites with CAL ≥ 6 mm (not on the same tooth) and ≥1 interproximal site with PD ≥ 5 mm were present. A moderate periodontitis was diagnosed if at least ≥2 interproximal sites with CAL ≥ 4 mm (not on same tooth) or ≥2 interproximal sites with PD ≥ 5 mm (not on one tooth) were present. If no severe or moderate periodontitis was present, periodontitis was diagnosed as mild or absent. In addition to the CDC classification, the new classification of 2017 [27] was applied. As no X-ray images were available for radiographically determining the bone loss index (bone loss (%)/age) for ethical reasons, only the stages of periodontitis were determined. 

### 2.4. Skin Autofluorescence (sAF)

sAF was measured using the validated sAF reader (AGE reader, DaignOptics, Groningen, The Netherlands) according to the method previously described [28]. The measurements were performed at a healthy pale skin site of the volar lower arm. The procedure was carried out three times and the mean value was calculated. sAF was indicated in arbitrary units (a.u.).

### 2.5. Determination of Peripheral Soluble Receptor of AGE (sRAGE) Concentration

sRAGE serum concentration was determined using a commercial kit (Quantikine^®^ ELISA Human RAGE Immunoassay for the quantitative determination of the extracellular domain of RAGE concentration, R&D Systems, Inc., Minneapolis, MN, USA) and according to the manufacturer’s protocol. Both cRAGE and esRAGE were determined and the combination reflects sRAGE. Measurements were performed in duplicate, and the results were averaged. 

### 2.6. Follow-Up

Follow-up data were collected from 95 patients. The follow-up was primarily carried out by telephone interview more than 3 years after CABG surgery. If follow-up information could not be obtained, we contacted civil registration offices and requested information about the patient’s current address or date of death. The categorical outcome parameters for this study were oriented on the known major adverse cardiac and cerebrovascular events (MACCE) criteria [29] in cardiac surgery: (1) no event; (2) myocardial infarction; (3) low cardiac output syndrome; (4) ventricular tachycardia (VT); (5) angina pectoris; (6) renewed revascularization surgery; (7) cardiac decompensation; (8) peripheral circulatory failure; (9) stroke/transient ischemic attack (TIA)/prolonged reversible ischemic neurological deficit (PRIND); (10) cardiac death; and (11) stroke death.

### 2.7. Statistics

#### 2.7.1. Power Analysis

To estimate the minimum size of the study group, a power analysis for Log Rank test for equality of survival curves was carried out using the software nQuery Advisor v.4.0, Statistical Solutions, Saugus, MA, USA. Our main hypothesis was that patients with severe periodontitis had an increased hazard ratio (HR) for the incidence of new cardiovascular events 3 years after CABG when compared with patients who had no severe periodontitis. Given a significance level of *p* = 0.05 and a power (1 − β) of 80%, 41 patients with severe and 41 individuals who do not have severe periodontitis needed to be included for a hypothetical HR of 2.0. 

#### 2.7.2. Statistical Procedures

Statistical analyses were carried out using a commercial software (SPSS v.25.0 package, IBM, Chicago, IL, USA). Values of *p* < 0.05 were considered as significant. Categorical variables were documented as number and the corresponding percentage in brackets. For comparisons, the chi-squared test was employed. If the expected values in one group were <5, Fisher’s exact test was performed. Metric demographic, clinical, and serological data were checked for normal distribution using the Kolmogorov–Smirnov test and the Shapiro–Wilk test. As none of the metric values were normally distributed, they were plotted as median and 25th/75th percentiles. For statistical evaluation, the Mann–Whitney U test was used. Correlations between two variables were calculated using the Spearman correlation test. For multivariate cross-sectional analyses, binary logistic regression was applied. Univariate survival analyses were carried out with Kaplan–Meier curves and Log-Rank test. Cox regression was applied in order to generate adjusted HRs. A receiver-operating characteristic (ROC) analysis including determining the Youden-index was carried out in order to measure the optimal threshold values for sAF and sRAGE to distinguish patients with and without a cardiovascular endpoint after CABG surgery. 

## 3. Results

### 3.1. Baseline Characteristics of Study Participants

The study participants had a median age of 69 years at the time of initial examination. The proportion of men was much higher than that of women. Over 60% of patients were current or past smokers. For most patients, three coronary vessels were affected by stenosis. Interestingly, all patients had at least a moderate periodontitis according to CDC classification or stage II according to the new classification. There was no patient without periodontitis. The median value for sAF was 2.7 (a.u.) and for sRAGE 568 pg/mL. Other important anamnestic and clinical data are summarized in Table 1. 

### 3.2. RAGE and sRAGE in Association to Severity of Periodontitis

According to the CDC classification, sAF was significantly (*p* = 0.006) increased in patients with severe periodontitis (2.8 a.u.) in comparison to individuals with moderate periodontitis (2.6 a.u.) (Figure 2a). If the relationship between sAF and the periodontitis staging according to the new classification was examined, a clear but not significant (*p* = 0.060) trend for a positive association between both parameters could be shown (Figure 2b).

sRAGE was slightly increased (567.9 vs. 574.5 pg/mL) in patients with severe periodontitis (Figure 3a). When using the new periodontitis classification (Figure 3b), sRAGE was inversely, but not significantly, associated with staging. 

### 3.3. RAGE and sRAGE Associated with BOP

BOP is a widely accepted factor for inflammation of subgingival periodontal tissue and a predictive factor for disease progression. Therefore, we wanted to investigate whether BOP correlated with sAF and/or sRAGE. sRAGE significantly negatively correlated with BOP but sAF did not (Table 2).

### 3.4. Multivariate Analysis

We then used binary logistic regression to investigate whether sAF and sRAGE represent independent risk indicators for severe periodontitis. This could be confirmed for sAF. The inverse association between sRAGE and severe periodontitis was significant, but the odds ratio (OR) was very low; thus, there is probably no clinical relevance here (Table 3). 

### 3.5. RAGE, sRAGE, and Severe Periodontitis Associated with New Adverse Cardiovascular Events after CAGB

We then investigated whether sAF, sRAGE, and severe periodontitis represent risk factors for cardiovascular outcome after CABG.

#### 3.5.1. Univariate Kaplan–Meier Analyses 

First, for sAF and sRAGE, the optimal thresholds for distinguishing the groups with event or without event after CABG were determined by applying a ROC analysis and calculating the Youden index. The threshold for sAF was 2.95 a.u. and for sRAGE at 599.3 pg/mL. According to these thresholds, both variables were dichotomized for Kaplan–Meier analyses. sAF (HR = 1.95, 95% CI = 1.1–3.61, Figure 4a) and sAGE serum levels (HR = 1.8, 95% CI = 0.86–3.76, Figure 4b) were significantly associated with a poorer cardiovascular outcome. Patients with severe periodontitis showed a striking trend, but it did not reach the level of statistical significance (HR = 1.8, 95% CI = 0.9–3.8) for a higher incidence of new cardiovascular events (Figure 4c).

#### 3.5.2. Multivariate Analysis with Cox Regression

The tested cofactors sAF and periodontitis tended to show an increased HR for new cardiovascular events after CABG. As the two factors did not reach statistical significance, however, they are not independent prognostic factors for cardiovascular outcome. We found no clinically relevant association between sRAGE and incidence of new cardiovascular events. Only previous MI and PAD were confirmed as independent risk factors for adverse events after CABG surgery (Table 4).

## 4. Discussion

Epidemiological studies have provided evidence for a positive association between periodontitis and incidence of CVD [30,31,32,33,34]. Moreover, the level of certain inflammatory mediators such as C-reactive protein and interleukin (IL)-6 were found to be associated with both diseases [35,36,37,38]. Obviously, inflammatory markers could be a link between periodontitis and CVD. In this context, the new biomarkers AGE and sRAGE seem to be important. Therefore, we examined whether these biomarkers were associated with the severity of periodontitis and whether they were predictors for cardiovascular outcome 3 years after CABG. Furthermore, we assessed whether the degree of periodontitis severity was also associated with the cardiovascular outcome. 

Regarding periodontitis, we found a significant association between tissue accumulation of AGEs measured as sAF and occurrence of severe periodontitis (CDC classification) in both univariate (Figure 2a) and multivariate analyses (Table 3). Thus, sAF may be an independent risk indicator for prevalence of severe periodontitis. sRAGE was negatively correlated with inflammation of subgingival periodontal tissue assessed as percentage of sites with BOP. Regarding outcome after CABG, both sAF and sRAGE were significantly associated with the incidence of new adverse events only in univariate Kaplan–Meier analyses (Figure 4a,b). For severe periodontitis, only a trend for an increased HR was shown (Figure 4c). In the multivariate Cox regression, sAF and sRAGE were no longer significant. Thus, all putative risk factors tested in this study were not independent risk factors for CABG outcome (Table 4). 

Our first hypothesis was that severe periodontitis is associated with sAF and sRAGE. This hypothesis could be confirmed for sAF because sAF was identified as an independent risk indicator for severe periodontitis (Table 2). To our knowledge, only one previous study has investigated a putative relationship between sAF and periodontal conditions. That study, conducted among young adults who suffered from type 1 diabetes, found a positive association between sAF and gingival inflammation, assessed as sulcus bleeding index (SBI). Further periodontal parameters such as CAL or PD, however, were not investigated. The authors conclude that sAF may be an indicator of early microvascular changes in gingival tissues appearing before any other periodontal conditions are visible [39]. Whether it is actually useful to measure sAF in routine dental diagnostic procedures must be clarified in further studies. For instance, in a cross-sectional study, generally healthy individuals with and without periodontitis should be compared with each other with regard to sAF. In another longitudinal study on subjects without periodontitis, it should be examined whether patients with elevated sAF values are actually more likely to develop periodontitis.

Our results indicate that sRAGE is likely to inhibit subgingival inflammation measured as BOP (Table 2). This supports the result of a previous study [21] that also showed an inverse association between BOP and sRAGE. This result may support the hypothesis that sRAGE isoforms may capture and eliminate circulating AGEs, decrease AGE serum levels, and in this way act as a competitive inhibitor for AGE-RAGE interaction and subsequent inflammatory reactions.

Our second hypothesis was that sAF, sRAGE, and severe periodontitis were putative risk indicators for new cardiovascular events within patients undergoing CABG. This hypothesis could only be partially confirmed for sAF and sRAGE in the univariate survival analysis (Figure 4a,b) because statistical significance was not reached after stratification for further confounders for CVD (Table 4). It is likely that the significant confounders “previous MI” and “PAD” used in the multivariate analysis mask the effect of sAF and sRAGE on CABG outcome.

The association between sAF and outcome after CABG was investigated in an another study [14] in which patients undergoing CABG, aortic valve replacement, or a combined procedure were analyzed. In contrast to our results, sAF proved to be an independent risk indicator for postoperative morbidity (OR = 2.8, *p* < 0.0001) and mortality (OR = 3.1, *p* < 0.0001) in that study. Two other studies showed in cardiac rehabilitation patients that sAF was independently associated with reduced exercise capacity [40] and in the case of patients with heart failures with new adverse cardiovascular events (MACE) [41]. Summarizing the available literature, it can be concluded that the noninvasive determination of sAF in cardiac surgery patients can be helpful in predicting cardiovascular outcome.

As already reported, sRAGE was identified as a predictor of postoperative outcome after CABG in two other studies [17,18]. This supports the theory that sRAGE reflects the activation of the AGE/RAGE system and ongoing inflammation. Two studies showed that circulating AGE correlated with sRAGE in both patients with type II diabetes [42] and nondiabetic individuals [43]. It was concluded that AGEs upregulate tissue RAGE expression. sRAGE could be generated from the cleavage of cell surface RAGE. Moreover, the authors reported that sRAGE may not efficiently capture and eliminate circulating AGEs [44]. Due to the still insufficient number of studies, further work is necessary to be able to estimate the prognostic value of determining sRAGE levels.

For severe periodontitis, we demonstrated that the HR for new events (HR = 1.8, *p* = 0.115, Figure 4c) tended to increase after CABG. Compared to the 1-year follow-up result [7], the HR was only slightly increased (HR = 1.6, *p* = 0.448). Nevertheless, we could not prove that periodontitis is a risk factor for patients who need CABG. So far, only a few studies have investigated a putative association between periodontitis and recurrent cardiovascular events. One study on 106 stroke/transient ischemic attack patients revealed that severe periodontitis was associated with an adjusted risk (HR = 2.8, *p* = 0.02) for new adverse events such as TIA, myocardial infarction, and vascular death [45]. A second study including 884 survivors of MI demonstrated only among never-smokers that mean CAL was associated with recurrent fatal and nonfatal cardiovascular events (HR 1.4 pro 1 mm CAL) after an average follow-up of 2.9 years [46]. A third study investigated 165 consecutive subjects with acute coronary syndrome (ACS) and 159 medically healthy, matched control subjects regarding periodontal conditions. After an observation period of 3 years, a positive association between alveolar bone loss caused by periodontitis and future ACS events was shown [47]. To our knowledge, there is still no study identifying periodontitis as a risk factor for recurrent cardiovascular events after CABG. Before any further conclusions can be drawn, however, a larger study is necessary.

### 4.1. Limitations of the Study

When interpreting the results of this study, two limitations should be noted. Firstly, none of the CVD patients included was periodontally healthy. This is due to the fact that in the age group examined (60 to 75 years), only very few individuals do not have periodontitis. In an earlier study, we revealed that among 1002 CVD patients of this age group, only 2.3% of individuals had no periodontitis [48]. A comparison between CVD patients with and without periodontitis regarding their sAF and sRAGE values and outcome after CABG may provide clearer results. Secondly, in the power analysis, we assumed for patients with severe periodontitis a hypothetical crude HR of 2.0 for new cardiovascular events. In fact, we achieved a HR of 1.8, which means that the target power of 80 was not reached. 

### 4.2. Conclusions

AGE measured by sAF was identified as an independent risk indicator for severe periodontitis.

## Figures and Tables

**Figure 1 jcm-11-04105-f001:**
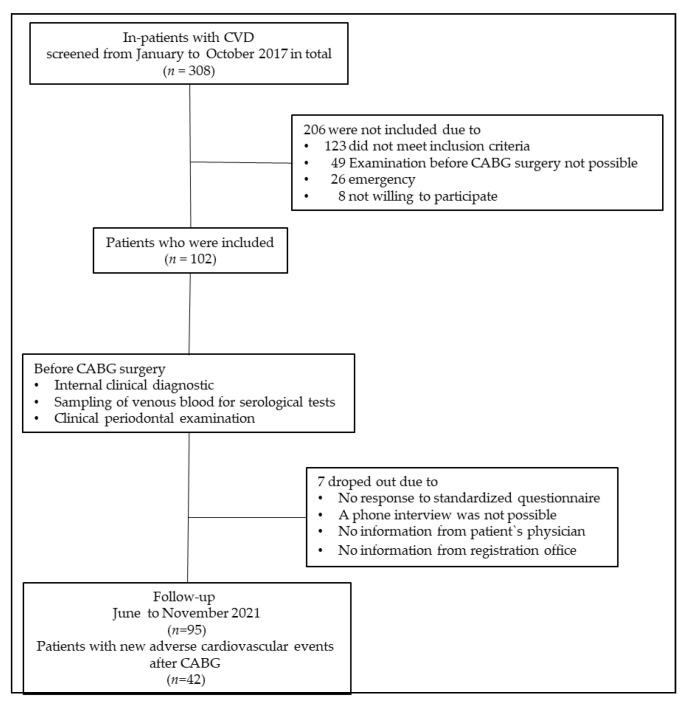
Study design and flow. CVD, cardiovascular disease; CABG, coronary artery bypass grafting.

**Figure 2 jcm-11-04105-f002:**
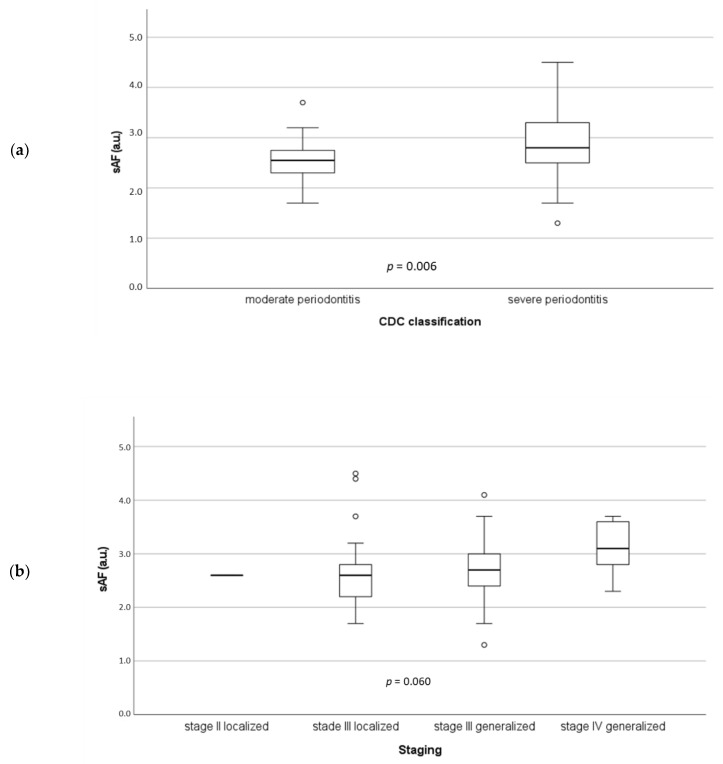
Skin autofluorescence (sAF) in related to periodontitis severity according to CDC classification (**a**) and new classification (**b**). o, outlier.

**Figure 3 jcm-11-04105-f003:**
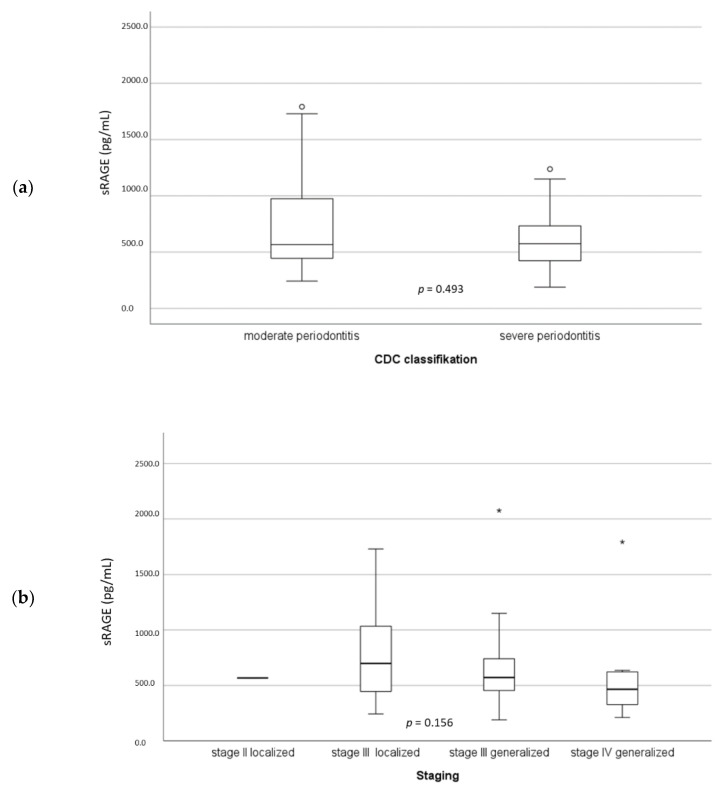
Level of soluble receptor of advanced glycation end products (sRAGEs) related to periodontitis severity according to CDC (**a**) and new classification (**b**). o, outlier; *, extreme value.

**Figure 4 jcm-11-04105-f004:**
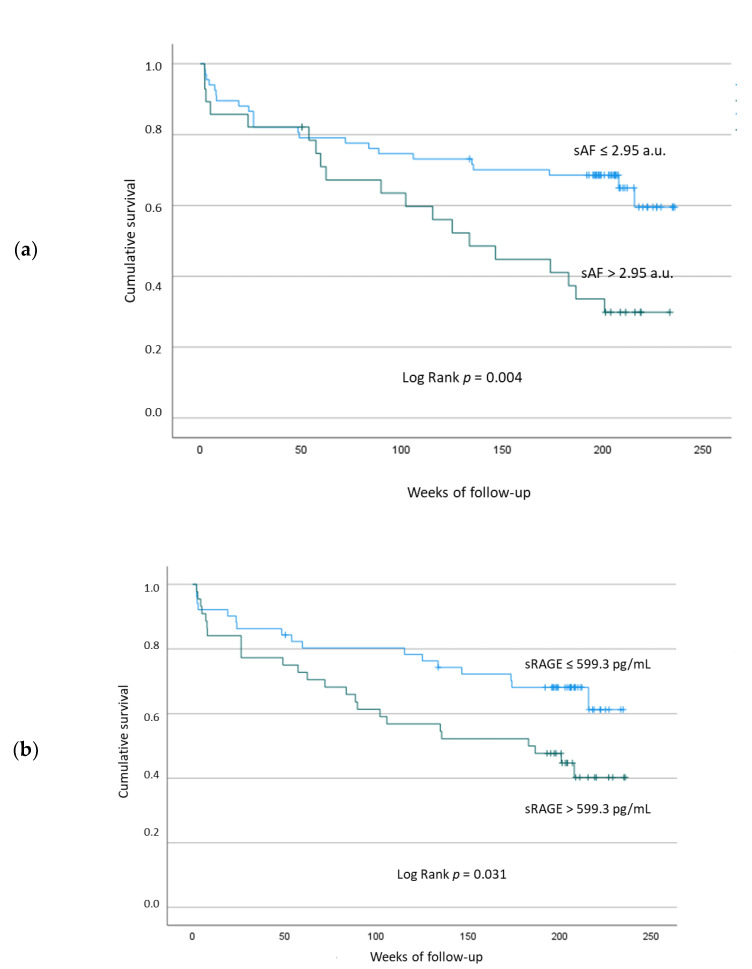
Kaplan–Meier-curves to investigate the influence of skin autofluorescence (sAF) (**a**), soluble receptor of advanced glycation end products (sRAGEs) (**b**) and severe periodontitis (**c**) on incidence of new cardiovascular events.

**Table 1 jcm-11-04105-t001:** Demographic and clinical variables of patients with cardiovascular disease (CVD).

Variable	Entire Study Cohort, *n* = 95 Median (25th/75th Percentile) or *n* (%)
Age (years)	69.0 (60.8/75.0)
Males	83 (88.3)
BMI (kg/m^2^)	28.7 (25.6/31.0)
Smoking	
Current	19 (20.2)
Past	40 (42.6)
Never	35 (37.2)
Affected coronaries	
One-vessel disease	5 (5.3)
Two-vessel disease	20 (21.3)
Three-vessel disease	68 (72.3)
More than three vessels affected	1 (1.1)
LVEF (%)	60 (45/60)
History of	
Diabetes mellitus	38 (40.4)
Hypertension	83 (88.3)
Dyslipoproteinemia	77 (81.9)
Peripheral arterial dis.	15 (16.0)
CVD	34 (36.2)
Myocardial infarction	26 (27.7)
Stroke/TIA	8 (8.5)
Angina pectoris	70 (74.5)
PTCA/stent	11 (11.7)
Atrial fibrillation	14 (14.9)
Periodontal conditions	
Plaque index (%)	1.3 (0.9/1.7)
Bleeding index (%)	18.2 (9.9/33.3)
Pocket depth (mm)	3.0 (2.6/3.5)
Attachment loss (mm)	3.9 (3.2/4.9)
Missing teeth	6 (3/13.5)
CDC classification (2007)	
No or mild	0 (0.0)
Moderate	29 (28.7)
Severe	72 (71.3)
New classification (2017)	
Staging	
I	0
II (localized)	1 (1.1)
III (localized)	24 (25.5)
III (generalized)	56 (59.6)
IV (generalized)	13 (13.8)
Blood values	
Creatinine (µmol/L)	85 (74.8/100)
Urea (mmol/L)	5.7 (4.5/7.1)
HbA1_C_ (mmol/mol)	37.6 (31.2/44.4)
CRP (mg/L)	2.7 (1.2/6.8)
Leukocytes (Gpt/L)	7.5 (6.5/9.0)
Platelet (Gpt/L)	238.5 (192.5/268.8)
Triglycerides (mmol/L)	1.1 (0/1.55)
Glucose (mmol/L)	6.5 (5.3/9.19)
sRAGE (pg/mL)	567.9 (436/770)
AGE tissue concentration (sAF) (a.u.)	2.7 (2.4/3.1)
Drugs	
Lipid lowering drugs	83 (88.3)
Oral anticoagulants	11 (11.7)
Antiarrhythmics	2 (2.1)

CVD, coronary vascular disease; MI, myocardial infarction; PTCA, percutaneous transluminal coronary angioplasty; CDC, Centers of disease control and prevention; TIA, transient ischemic attack; HbA1c, glycated hemoglobin; CRP, C-reactive protein; sAF, skin autofluorescence; a.u., arbitrary units, sRAGE, soluble receptor of advanced glycation end products; LVEF, left ventricular ejection fraction.

**Table 2 jcm-11-04105-t002:** Correlation between bleeding on probing (BOP) and skin autofluorescence (sAF)/soluble receptor of advanced glycation end products (sRAGEs).

Variables		BOP	sAF	sRAGE
BOP	*r*	1.000	0.106	−0.189
	*p*		0.154	0.034
	*n*	95	95	94
sAF	*r*	0.106	1.000	0.086
	*p*	0.154		0.205
	*n*	95	95	94
sRAGE	*r*	−0.189	0.086	1.000
	*p*	0.034	0.205	
	*n*	94	94	95

*r*, Spearman’s rank correlation coefficient; *p*, Significance (one sided).

**Table 3 jcm-11-04105-t003:** Odds ratios (OR) for severe periodontitis (CDC classification) adjusted for sRAGE, sAF, and known confounders for severe periodontitis.

Confounding Variables	Odds Ratio	95% Lower	CI Upper	*p*-Values
Age	1.034	0.974	1.097	0.278
Gender (male)	1.467	0.328	6.558	0.616
Current smoking	1.483	0.492	4.776	0.484
Plaque index	1.566	0.664	3.696	0.306
Diabetes mellitus	1.119	0.398	3.146	0.831
sRAGE	0.999	0.997	1.000	0.049
sAF	2.891	1.119	7.466	0.028

sRAGE, soluble receptor of advanced glycation end products; sAF, skin autofluorescence; CI, confidence interval.

**Table 4 jcm-11-04105-t004:** Hazard ratios for new cardiovascular events after CABG within a 3-year follow-up adjusted for sRAGE, sAF, severe periodontitis (CDC classification), BOP, and other confounders for cardiovascular events.

Confounding Variables	Hazard Ratio	95% Lower	CI Upper	*p*-Values
Age	1.016	0.974	1.060	0.454
Gender (male)	0.678	0.224	2.051	0.492
Current smoking	0.899	0.391	2.065	0.801
Severe periodontitis (CDC)	1.701	0.737	3.930	0.214
Bleeding on probing (BOP)	0.993	0.978	1.008	0.345
Atrial fibrillation	2.411	0.931	6.245	0.075
PAD	2.724	1.330	5.578	0.006
Previous MI	2.797	1.281	6.107	0.010
Urea	1.084	0.918	1.280	0.342
Creatinine	1.005	0.987	1.023	0.593
sRAGE	1.000	0.999	1.001	0.920
AGE tissue concentration (sAF)	1.401	0.782	2.513	0.257

CDC, Centers of Disease control and Prevention; PAD, peripheral arterial disease; MI, myocardial infarction; sRAGE, soluble receptor of advanced glycation end products; sAF, skin autofluorescence; CI, confidence interval.

## Data Availability

Data can be provided only by the authors.

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
