# Peer review of "Advanced Glycation End Product (AGE) and Soluble Receptor of AGE (sRAGE) Levels in Relation to Periodontitis Severity and as Putative 3-Year Outcome Predictors in Patients Undergoing Coronary Artery Bypass Grafting (CABG)"

_jcm, 2022, doi:10.3390/jcm11144105_

Round 1
Reviewer 1 Report
see attached file

Author Response
"Please see the attachment."

Reviewer 2 Report
Thank you for submitting this manuscript for consideration.
The main concerns are:
- There is very little mechanistic evidence to link sRAGE/periodontitis/CABG - how are AGEs involved in the pathology of the clinical states of periodontitis and CABG?
- There is no sample size calculation and the correlations seen could very well be spurious
- It is unclear how the results presented is useful clinically
Author Response
"Please see the attachment."

Reviewer 3 Report
The manuscript evaluates the relationship between periodontitis and AGE / sRAGE in a color of patients eligible for coronary artery bypass grafting. The authors concluded that a trend for an increased Hazard Ratio for new cardiovascular events could be observed in patients affected by severe periodontitis.
The article is overall well-written and data clearly presented. Methodology is properly descrive. The authors are suggested to add sample size calculation in order to better assess the power of the analysis conducted.
Author Response
"Please see the attachment."

Reviewer 4 Report
This is a very interesting study, but the abstract and the discussion of the association between periodontitis and CABG are too illogical.
1.It is well known that AGEs and RAGE are involved in various diseases, especially in peripheral vascular diseases. Therefore, Table 3 is a natural result. Therefore, we conclude that cardiac surgery cannot be predicted by the severity of periodontitis.
2.Table 2 also did not examine the relationship between the severity of periodontitis and CABG. Therefore, this study should only conclude that there is an association between AGEs and RAGE and the severity of periodontitis.
Author Response
"Please see the attachment."

Round 2
Reviewer 1 Report
This version is improved from the previous version. However, the reviewer do not see the importance of sAF as a diagnostic tool for periodontal diagnostics considering the procedure is a routine in the dentist visit and the sAF reader is not common in the dentist office. In addition, the rationale of using AGEs, sRAGE and periodontists to predict the outcome of CABG is not well established. Finally, the quality of presentation needs significant improvement.
Author Response
Reviewer 1
This version is improved from the previous version. However, the reviewer do not see the importance of sAF as a diagnostic tool for periodontal diagnostics considering the procedure is a routine in the dentist visit and the sAF reader is not common in the dentist office. In addition, the rationale of using AGEs, sRAGE and periodontists to predict the outcome of CABG is not well established. Finally, the quality of presentation needs significant improvement.
Answer: The reviewer is right that clinical symptoms such as probing depth, loss of attachment and alveolar bone visible in the X-ray are decisive in the diagnosis of periodontitis. However, all of these conditions are the result of a disease process that has already run its course. Therefore, researchers are looking for biomarkers that are suitable for early diagnosis, especially of severe disease progression, in order to be able to offer affected patients an individual therapy. Whether the measurement of sAF is actually useful in routine dental diagnostics must be clarified in further studies. For this purpose, for instance generally healthy individuals with and without periodontitis have to be compared with each other with regard to sAF. This has been discussed in the revised manuscript.
Furthermore, the biological background of the relationship between periodontitis and CVD is not yet fully understood. Here, one assumes direct or indirect damage to the coronaries by periodontal bacteria. Furthermore, it is known that severe periodontitis increases the proinflammatory status, e.g. level for CRP and IL-6 which could contribute on pathogenesis of CVD. In this context, it was interesting to investigate whether periodontitis is also associated with the level of sRAGE or sAF. Both markers have been identified in previous papers (see numbers 17, 18, 14, 40, 41 in the reference list) as prognostic factors for CABG outcome. In particular, the determination of sAF as a non-invasive method is therefore likely to be of particular importance in early cardiac diagnostics. This has been discussed in the revised manuscript.
A native speaker with experience in editing medical texts revised the entire manuscript.
Reviewer 2 Report
One of the aims of this study is to investigate whether sAF and sRAGE are independent risk factors of severe periodontitis. But since you are only looking at patients who already have periodontitis, how do you know whether the periodontitis caused high levels of sRAGE, or vice versa?
Same comments go for risk of cardiovascular events after CABG and sRAGE
